# How Our Microbiome Influences the Pathogenesis of Alopecia Areata

**DOI:** 10.3390/genes13101860

**Published:** 2022-10-14

**Authors:** Pedro Sánchez-Pellicer, Laura Navarro-Moratalla, Eva Núñez-Delegido, Juan Agüera-Santos, Vicente Navarro-López

**Affiliations:** 1MiBioPath Research Group, Department of Clinical Medicine, Health Sciences Faculty, Catholic University of Murcia, Campus de los Jerónimos 135, 30107 Murcia, Spain; 2Infectious Diseases Unit, University Hospital of Vinalopó-Fisabio, Carrer Tonico Sansano Mora 14, 03293 Elche, Spain

**Keywords:** alopecia areata, non-scarring alopecia, skin microbiota, hair follicle microbiota, scalp microbiota, gut microbiota, probiotics, pathogenesis, immune privilege collapse

## Abstract

Alopecia areata is a multifactorial autoimmune-based disease with a complex pathogenesis. As in all autoimmune diseases, genetic predisposition is key. The collapse of the immune privilege of the hair follicle leading to scalp loss is a major pathogenic event in alopecia areata. The microbiota considered a bacterial ecosystem located in a specific area of the human body could somehow influence the pathogenesis of alopecia areata, as it occurs in other autoimmune diseases. Moreover, the Next Generation Sequencing of the 16S rRNA bacterial gene and the metagenomic methodology have provided an excellent characterization of the microbiota. The aim of this narrative review is to examine the published literature on the cutaneous and intestinal microbiota in alopecia areata to be able to establish a pathogenic link. In this review, we summarize the influence of the microbiota on the development of alopecia areata. We first introduce the general pathogenic mechanisms that cause alopecia areata to understand the influence that the microbiota may exert and then we summarize the studies that have been carried out on what type of gut and skin microbiota is found in patients with this disease.

## 1. Introduction

Alopecia areata is a complex multifactorial autoimmune-based disease in which genetic predisposition is also especially important. Therefore, immunity and genetics are the factors contributing the most to the onset and development of this disease [1].

Alopecia areata is the second leading cause of non-scarring alopecia after androgenic alopecia [2]. The fact that it is non-scarring implies that there is preservation and not destruction of the hair follicle. The clinical development of this type of alopecia is quite variable and unpredictable. The most ordinary form of presentation is the appearance of well-defined patches of scalp or body hair loss (beard, e.g.,) without signs of inflammation [3]. The development of alopecia areata may be self-limiting (hair regrow) or the patient may experience periods of unpredictable remissions and recurrences [4]. In some patients, it may even establish itself as persistent [5].

Several subtypes of alopecia areata have been described and are related to the prognosis of the disease [4]. As mentioned, patchy alopecia is the most frequent subtype. More severe phenotypes are alopecia totalis, where there is a total loss of scalp, and alopecia universalis, where there is a total loss of body hair and scalp. Another severe phenotype since it is quite refractory to treatment and has a poor prognosis is ophiasis, where the pattern of alopecia usually appears in a band in the area from the top of the ears downwards [3]. In some cases, alopecia areata can start with a well-defined patch and evolves into alopecia totalis, alopecia universalis or ophiasis [5]. Sisaipho is a subtype with only extensive scalp loss in the central part. Acute diffuse and total alopecia is a very rapid diffuse hair loss over the scalp which has a good prognosis and spontaneous resolution [6]. In Marie Antoniette’s syndrome in women and Thomas More’s syndrome in men, there is a sudden loss of pigmented hair on the scalp with the preservation of white hair [7].

Other factors influence the prognosis of alopecia areata and its development. The persistence of hair loss is a key prognostic factor. The prognosis is worse when alopecia areata persists beyond 5 years and hair loss can become irreversible when it persists for 10 years [8]. Moreover, early age of onset with a severe phenotype at disease onset (extent and duration of hair loss) is a poorer prognostic factor [3]. Moreover, hair loss in regions other than the scalp, such as eyebrows and eyelashes, may indicate a poor prognosis [8]. Furthermore, nail changes such as pitting and trachyonychia are signs associated with severe forms of the disease such as alopecia totalis or alopecia universalis, and therefore, their development is associated with a worse prognosis [9]. These nail changes usually occur in about 30% of cases, although there is a lot of variability in studies [10], and they are also more common in cases of alopecia areata in children than in adults [10].

Several studies have estimated the prevalence and incidence of alopecia areata; however, few studies have examined the epidemiology of the different subtypes such as alopecia totalis, alopecia universalis and ophiasis. In a recent systematic review and meta-analysis of observational studies, it has been estimated that the overall prevalence of alopecia areata is around 2%. Furthermore, this study has observed that there is significant regional variability, that alopecia universalis, alopecia totalis and ophiasis affect less than 1 in 1000 people, and that there has been a significant increase in recent years in both alopecia areata and these subtypes [11]. In addition, no differences were observed in the prevalence of alopecia areata according to sex, and the prevalence was significantly higher in studies of children and adolescents than in studies of adults [11]. Another systematic review published in 2015 has established that alopecia areata is more prevalent in young people. The age groups affected in order are firstly those aged 21–40 years followed by those aged 1–20 years, 41–60 years, and lastly, those aged 61–80 years [12]. Moreover, this systematic review establishes the average age of onset of alopecia areata as between 25 and 36 years old, although in children (early-onset alopecia areata) it ranges from 5 to 10 years old, usually with more severe subtypes. [12].

Another important epidemiological feature of alopecia areata is that it is associated with several comorbidities. A recent systematic review published in 2019 showed that patients with alopecia areata have a higher prevalence of atopic dermatitis and allergic rhinitis, *Helicobacter pylori* infection, hyperinsulinemia, metabolic syndrome, lupus erythematosus, iron deficiency anaemia, autoimmune thyroid diseases, and psychiatric diseases such as anxiety, depression, obsessive-compulsive disorder and alexthymia [13]. In addition, these patients have a higher risk of developing systemic lupus erythematosus, psoriasis, and rheumatoid arthritis, all of them autoimmune diseases [13]. In this regard, the recent clinical guideline of the Italian Society of Dermatology recommends testing for at least specific autoantibodies for autoimmune thyroiditis and celiac disease [14].

The treatment of alopecia areata is a complicated issue. There are several therapeutic options available, which are supported by little clinical evidence through high-quality randomized clinical trials. The only systematic review published in Cochrane in 2008 concluded that there is no quality evidence that any treatment from clinical trials provides long-term beneficial effects in patients with alopecia areata and in the subtypes alopecia totalis and alopecia universalis [15]. Furthermore, only one study of this review comparing two topical corticosteroids showed short-term beneficial effects [16]. In relation to this, it could be pragmatically assumed that in clinical practice, mild cases of alopecia areata are treated with topical or intralesional corticosteroids, and moderate to severe cases with systemic corticosteroids, immunosuppressive agents, janus kinase (JAK) inhibitors, different types of immunotherapy and biological drugs, minoxidil, laser therapy, etc. [4,17]. Nevertheless, a recent meta-analysis has reported that pentoxifylline plus topical corticosteroids has the highest success rate, although other options such as intralesional corticosteroids may also be valid. It also rules out the efficacy of some treatments such as JAK inhibitors, azathioprine, cyclosporine, or tacrolimus [18]. This is important because clinical guidelines do not recommend treatment, but rather set out a list of possibilities, and if they make recommendations, they are usually based on expert opinion [14]. As mentioned earlier, the problem with alopecia areata patients is that they exhibit an unpredictable course [4]. It is possible that the disease may recur after cessation of treatment [5]. In addition, the long-term effects are poorly understood [5]. It is noteworthy that many treatments have potential adverse effects [4]. Therefore, the assessment of establishing a treatment for alopecia areata is complicated. However, research for more effective treatment of alopecia areata continues. In this regard, Baricitinib has recently been approved in both Europe and the United States. It is an oral inhibitor of JAK 1 and 2 that can interrupt the signaling of cytokines involved in the pathogenesis of alopecia areata [19]. Moreover, mini-pulse therapy with oral corticosteroids could be safer in maintaining efficacy in cases of severe alopecia areata [20].

The pathogenesis of alopecia areata has not been fully elucidated [21], although there is evidence that it is an autoimmune disorder [22] that is also influenced by genetic predisposition [1] and other factors [23]. Due to the relationship of the skin and gut microbiota with other systemic autoimmune diseases [24] and other immune-mediated skin diseases [25,26,27], several studies on the microbiota of alopecia areata patients have been carried out in recent years [28].

The aim of this narrative review is to examine the published literature on the cutaneous and intestinal microbiota in alopecia areata in order to be able to establish a pathogenic link. In this review, we summarize the influence of the microbiota on the development of alopecia areata. We first introduce the general pathogenic mechanisms that cause alopecia areata to understand the influence that the microbiota may exert and then we summarize the studies that have been carried out on what type of gut and skin microbiota is found in patients with this disease.

## 2. Pathogenesis of Alopecia Areata

### 2.1. Hair Cycle Disruption

The pilosebaceous unit is composed of the hair follicle in close relation to the sebaceous gland together with the pili muscles and the capillaries, which supplement it in its lower part through the dermal papilla. The infundibulum is the section of the mature hair follicle that begins with the projection into the epidermis until the entry of the sebaceous gland. Next, in the dermis, the bulge is situated, a region of vital importance in the hair cycle as it harbors the stem cells that will generate the transit-amplifying cells during the growth phase. The bulb is located (a regeneration area) in a region deeper than the bulge. The bulb is composed of the hair follicle matrix, surrounding the dermal papilla. In the hair follicle matrix, there are melanocytes, which produce the hair pigment. In the suprabulbar area the hair follicle epithelium is located, composed of concentric layers forming the outermost outer root sheath, the inner root sheath, and the most internal hair shaft (it will be the visible part of the emerging hair) [29].

Basically, the hair cycle alternates phases of growth, involution, and quiescence. In the active growth or anagen phase, matrix cells proliferate and differentiate into transit-amplifying cells, which mobilize to the bulb and hair shaft. As the nutrition supply to the matrix cells diminishes and the catagen phase is beginning, the lower hair follicle regresses and the hair shaft growth ceases. Moreover, apoptosis is now occurring in the lower hair follicle which causes the separation of the dermal papilla (and blood vessel). Afterwards, the hair follicle enters the quiescent or telogen phase. However, the telogen phase is highly dynamic since the necessary conditions are being generated to start a new cycle with a new anagen phase. When the telogen phase is ending, the hair shaft is shed so that a new hair shaft can begin to grow [30].

Each of the different phases of the hair follicle cycle is regulated by signal molecules and cells involved in complex signaling pathways. When this homeostatic balance is lost, the development of the hair follicle cycle is affected [31]. In alopecia areata patients, different hair cycle disorders are reported. In the acute phase of the disease, peribulbar and intrabulbar inflammatory infiltration are characteristic, although it may not be observed in chronic stages [32]. The infiltrate comprises mainly T-lymphocytes, Langerhans cells, histiocytes, plasma cells, mast cells and eosinophils. T-cells cause apoptosis of the more peripheral matrix cells resulting in thinning of the hair shaft and the appearance of the typical exclamation mark hairs. This inflammation also causes a rapid transition from the anagen phase to the catagen and telogen phases, which leads to a reversal of the anagen/telogen ratio in alopecia areata patients. It is important to highlight that the inflammatory infiltrate does not affect the stem cells, so the hair follicle has the capacity to regenerate if it disappears after a new growth cycle. However, if during the new anagen phase, the hair follicle is once again affected by peribulbar and intrabulbar inflammation, the cycle will be interrupted prematurely and alopecia areata will persist [33].

### 2.2. Hair Follicle Immune Privilege Collapse and Autoimmune Response

The hair follicle presents the characteristic of being a region of immune privilege (IP), for instance certain organs such as the brain, eyes, gonads, and placenta [34]. This was firstly hypothesized by Paus et al., in 2003 [35]. The IP is the main mechanism by which tolerance is induced and immune recognition is prevented due to the presence of potential autoantigens in the aforementioned tissues. In the hair follicle, this protects the stem cells and derived cells that constitute its epithelium [36]. There are several forms to maintain the hair follicle with IP in normal conditions. As a physical barrier, an extracellular matrix protects the hair bulb by preventing the infiltration of immune cells [22]. A reduction in the expression of major histocompatibility complex (MHC) class I molecules is observed due to the local production of immunosuppressive factors such as α-melanocyte-stimulating hormone (α-MSH), transforming growth factor-β (TGF-β) and indoleamine-2,3-dioxygenase (IDO) among others [37]. There is also a decrease in the expression of MHC class II molecules on Langerhans cells, which are a type of dendritic cells in the skin, thus there is an impairing of the antigen-presenting function [38]. Moreover, the hair bulb (and bulge, so IP is not only restricted to the bulb) shows a very high expression of the transmembrane glycoprotein CD200, which interacts with its receptor on several immune cells to generate tolerance signals, decrease the activity of antigen-presenting cells (APCs) and decrease the secretion of pro-inflammatory cytokines [39,40]. Moreover, during the crash of the IP there is a secretion of chemokines such as IL-15, IL-2 and various CXCLs (CXC ligands), which act as chemotactics for other immune cells in the hair follicle [38].

The collapse of the IP of the hair follicle is the main condition for the initiation of the mechanisms that cause hair loss in alopecia areata. Therefore, it leads to an overexposure of autoantigens during the anagen phase (mainly), and the hair follicle cycle is corrupted, which explains the dermoscopic findings characteristic of alopecia areata [38]. Some authors defend the hypothesis that the hair follicle autoantigens are related to certain pigments produced by melanocytes (or derived products), which would be immunogenic and due to this the hair follicle is established as a region with IP [41,42].

Two major immunological elements in the pathogenesis of alopecia areata are T-lymphocytes CD8+NKG2D+ and INF-γ (interferon γ). Under normal conditions with an IP in the hair follicle, there is a decreasing in the expression of the NK cell receptor D (NKGD2) and its ligands MICA (MHC class I chain-related A) and ULBPs (UL16-binding proteins). Upon IP collapse, NKG2D receptors in NK cells and T-cells CD8+, and its ligands in the epithelium of hair follicles are upregulated [43]. Xing et al., in 2014, demonstrated that cytotoxic T-cells CD8+NKG2D+ are necessary and sufficient for the establishment of alopecia areata in the susceptible disease murine model C3H/HeJ (principal, genetically susceptible, murine model of alopecia areata). Basically, T-cells CD8+NKG2D+ produce INF-γ through JAK1 and JAK3 signaling pathways, which in turn triggers hair follicle epithelial cells to produce IL-15, which interacts with T-cells CD8+NKG2D+ to produce more INF-γ through JAK1 and JAK2, thus establishing a positive feedback loop [44]. T-lymphocytes CD8+NKG2D+ are the major contributors to hair loss in alopecia areata and are the first cells to infiltrate the hair follicle [1]. These cytotoxic T-cells CD8+ attack the hair follicle through granzyme B, which is elevated in the hair follicle of humans with alopecia areata [45]. Similarly, high expression of migration inhibitory factor (MIF) prevents the infiltration of NK cells CD56+NKG2D+, but when the IP collapses, these cells infiltrate the hair follicle [46]. In this regard, Gilhar et al., in 2013, demonstrated the importance of the human peripheral blood mononuclear cells (PBMCs) enriched for NKG2D+ and CD56+ cells (including NK cells). They performed several normal hairy human scalp engraftments in 25 SCID mice (mice that are severely deficient in functional B and T-lymphocytes) and injected them with a cell suspension of PBMCs enriched for NKG2D+ and CD56+ cells. At 3–5 weeks they observed histological and dermoscopic findings of alopecia areata where the injections were administered in the scalp transfer. Thus, as in alopecia areata in humans, a peribulbar infiltrate of T-lymphocytes and a collapse of the IP were observed [47]. 

INF-γ breaks down the IP of the hair follicle and is a key player in the pathogenesis of alopecia areata. INF-γ is mainly produced by NK and NKT cells. In addition to being a key element in the vicious cycle between T-lymphocytes CD8+NKGD2+ and hair follicle epithelial cells with the intervention of the intracellular JAK signaling pathway system, INF-γ induces the expression of MHC molecules, NKGD2 and its ligands in the hair bulb during the anagen phase, thus promoting autoimmunity against the hair follicle [22]. In this context, Gilhar et al., in 2005, observed that INF-γ accelerated the development of alopecia areata in C3H/HeJ mice. This was associated with increased expression of MHC class I and II molecules in the hair follicle epithelium [48]. Moreover, Freyschmidt-Paul et al., in 2006, engrafted skin with alopecia areata from C3H/HeJ mice into other C3H/HeJ mice with (knockout-type) and without (wild-type) deletion in the gene encoding INF-γ. While 90% of the wild-type mice developed alopecia areata, none of the knockout mice developed hair loss. Thus, knockout mice fail to activate Th1 cells in response to transplanted autoantigens, implicating the key role of INF-γ in the induction of alopecia areata [49].

Plasmacytoid dendritic cells (PDCs) have become of recent interest in the pathogenesis of alopecia areata. Ito et al., in 2020, observed that INF-α produced by PDCs cause alopecia areata in C3H/HeJ mice by stimulating apoptosis and increasing the secretion of chemokines such as CXCL10, infiltrating Th1/Thc1 cells [50]. PDCs are not found in healthy skin; however, they can infiltrate when an infectious or autoimmune pathology is established, such as viral infections or psoriasis [51]. PDCs produce large amounts of type I interferons such as INF-α or INF-β and have been shown to participate in the antiviral response. In fact, some studies have reported that alopecia areata can be established after a viral infection [52].

T-lymphocytes CD4+ Th17 and Treg are also involved in the pathogenesis of alopecia areata, as Th17 infiltrates the dermis and around the hair follicle (T-cells CD4+ were abundant after upregulation of MHC class II molecules in the hair follicle) [53]. Moreover, in patients with alopecia areata, there is a systemic imbalance between Treg and Th17 cells, a not uncommon finding in autoimmune diseases [54]. Moreover, Th17-derived cytokines are increased both in alopecia areata lesions and in the blood of alopecia areata patients [55].

Plasmatic TNF-α (tumor necrosis factor α) is also increased in patients with alopecia areata [56]. In alopecia areata lesions, TNF-α is derived from T-lymphocytes (CD4+ and CD8+) [57]. TNF-α acts by interfering with the hair follicle cycle, but its mechanism could be paradoxical since it could inhibit the overexpression of MHC class I molecules in the hair follicle, as well as suppress the development of PDCs responsible for the coordination of T-cells CD4+, CD8+ and NK cells in the establishment of alopecia areata [1].

Other elements that could be involved in the pathogenesis of alopecia areata are mast cells that can drive an antigen-specific cytotoxic T-cell CD8+ response in the hair follicle [58], neuropeptides (secreted by the sensory nerve fibers that innervate the hair follicle), and neurohormones such as substance P, vasoactive intestinal peptide (VIP), calcitonin gene-related peptide (CGRP), somatostatin, corticotropin-releasing hormone (CRH), adrenocorticotropic hormone (ACTH), cortisol, etc. [38]. It is beyond the scope of this review to discuss in detail these molecular mechanisms involved.

These multiple possible pathogenic immunological pathways involved in alopecia areata could explain the clinical heterogeneity observed in these patients with different developments such as spontaneous remission without relapses versus severe progressive evolution towards universal alopecia, as well as different associations with other diseases (comorbidities) and differences in response to different treatments.

### 2.3. Role of Genetics in Alopecia Areata

Observational studies have shown a genetic component in alopecia areata, demonstrating a high incidence in first-degree relatives and a high concordance in twins [59]. This implies that family history is a risk factor for alopecia areata. Nevertheless, there is variability in the prevalence and development of alopecia areata in patients with a family history and it is difficult to predict a pattern, suggesting that other factors such as environmental or lifestyle factors are necessary for alopecia areata to develop [1].

Alopecia areata is a complex polygenic disease with hundreds of single nucleotide polymorphisms (SNPs) found in patients. Many of these polymorphisms are found in genomic regions involved in the phenotype of the immune system, including the activation and functionality of Treg and cytotoxic T-lymphocytes, cytokine expression and antigen presentation [43]. Many of these genes are also associated with other autoimmune or immune-based diseases such as inflammatory bowel disease, multiple sclerosis, type 1 diabetes mellitus and psoriasis [60].

Several genes encoding HLA (human leukocyte antigen) variants have been correlated with alopecia areata, among them, polymorphisms in HLA-DRB1 are a major contributor to the disease phenotype. A recent meta-analysis has shown that HLA-DRB1*04 and HLA-DRB1*16 variants increase the risk of alopecia areata, while HLA-DRB1*0301, HLA-DRB1*09 and HLA-DRB1*13 variants are protective [61]. These polymorphisms tend to occur in the peptide-binding region of the APCs, thus affecting their binding affinity and therefore are related to autoimmune processes [62].

Regarding genes involved in hair follicle IP and hair follicle collapse that occurs in alopecia areata, the genes encoding NKG2D and its ligands such as MICA and ULBP are also relevant, and they are only involved in the pathogenesis of alopecia areata and not in other autoimmune diseases. Some polymorphisms in these genes may modify the receptor–ligand affinity or increase expression, which would lead to greater activation [63]. SNPs in the genes encoding some heat shock proteins (HSPs) [64] and genes involved in the elimination of reactive oxygen species (ROS) such as PRDX5 and ACOX have also been linked to alopecia areata [65]. It is noteworthy that some of these genes encode stress-inducible proteins or are associated with a greater effect of stress in alopecia areata [66], which will be discussed later in one of the next sections, which is another factor that influences the development of the disease.

As genes are involved in central and peripheral tolerance, there are several associated with alopecia areata. Polymorphisms in the autoimmune regulator gene (AIRE), whose complete loss of function produces polyendocrinopathy-candidiasis-ectodermal dystrophy (APECED) where alopecia areata is a common sign, continue to be associated with alopecia areata in patients without APECED [67]. Other genes implicated include forkhead box P3 (FOX3P) [68], ikaros family zinc finger 4 (IKZF4) [43], glycoprotein A repetitions predominant (GARP) [65], cytotoxic T lymphocyte-associated antigen 4 (CTL4) [43], IL-2A receptor chain (IL-2RA) [43], protein tyrosine phosphatase no receptor 22 (PTPN22) [69] and transporter associated with antigen processing 1 (TAP1) [70] among others [66].

SNPs in genes encoding cytokines involved in the progression of alopecia areata after the IP collapse are also associated. These genes encode IL-1 family, IL-4, IL-13, IL-17, IL-6, IL-26, IL-12B and TNF family [66]. Other genes are related to the progression of alopecia areata after the IP collapse of hair follicles such as the suppressor of cytokine signaling-1 (SOCS1) gene [43], FAS and FAS ligand genes [71], the neurogenic locus homolog 4 (NOTCH4) [72] and some chemokines encoding genes [73].

Likewise, some genes involved in the pigmentation process have also been associated with alopecia areata, such as the gene encoding the melanin-concentrating hormone receptor 2 (MHCR2) [74]. In addition, autophagy-related genes that in the context of alopecia areata act at the level of melanogenesis, autoimmunity and hair follicle cycling have also been linked, such as gene encoding syntaxin 17 (STX17) [43].

Finally, some alopecia areata-associated genes encode proteins involved in the hair follicle cycle that operate as structural proteins or transcription factors. These include, for example, the SMARCA2 gene [75].

### 2.4. Additional Influential Factors in the Development of Alopecia Areata

Oxidative stress could play a role in the pathogenesis of alopecia areata as it is a disease where an inflammatory condition is established. The ROS generated could exceed the cellular antioxidant capacity and this could influence the development of the disease. Biomarkers of oxidation/anti-oxidation status have provided inconsistent results in some studies [76,77]. In this context, a recent meta-analysis of 18 studies published in 2020 concluded that current evidence suggests that alopecia areata is associated with oxidative stress, but that further studies are required to provide robustness to this association. Furthermore, oxidative stress biomarkers are more increased in severe forms of the disease compared to mild or moderate forms [78]. Recently, Sachdeva et al., in 2022, found that total antioxidant status (TAS) and superoxide dismutase (SOD) levels were decreased and malondialdehyde (MDA) increased in 40 patients with alopecia areata compared to 40 age- and sex-matched healthy controls [79]. Furthermore, TAS and SOD decreased, and MDA increased with respect to disease severity. Therefore, the data suggest that oxidative stress seems to influence the development of alopecia in terms of its severity. On the other hand, in the previous section about genetics and alopecia areata, it has been mentioned that some associated polymorphisms also act at the oxidative stress level. Polymorphisms in stress-inducible protein genes such as HSPs [64] and NKGD2 receptor ligands [63], as well as in genes involved in ROS clearance such as PRDX5 and ACOX [65], have been linked to alopecia areata [66]. It is also noteworthy that ROS can occur in the context of situations other than alopecia areata as a consequence of treatment, infections or established comorbidities [80,81].

Although some patients who develop alopecia areata describe a previous episode of psychological stress, other studies have not found this event. So, it is certain that a close causal relationship between alopecia areata and emotional stress has never been established [1,82]. However, there is relative evidence that the central nervous system (CNS) activation in response to psychological stress influences the pathogenesis of alopecia areata to some degree. Much of this information has been obtained from murine models [82]. Firstly, it is important to remark that the hair follicle is a producer and target of hormones related to psychological stress. Thus, in the skin, there is a system analogous to the hypothalamic–pituitary–adrenal (HPA) axis that becomes activated during physical aggression or psychological stress in the same way as in the CNS [83]. Psychological stress activates both the HPA and the brain–hair follicle (BHF) axis. This could lead to some findings that would be involved in the pathogenesis of alopecia areata. Increased secretion of substance P by the sensory nerve fibers of the hair follicle causes perifollicular activation of mast cells, which leads to inhibition of hair follicle growth in the anagen phase [84], as well as causing cutaneous neuroinflammation [85]. Nerve growth factor (NGF), a neurotrophin that is increased during stress, also contributes to the disruption of the hair follicle cycle and the collapse of its IP [85]. Likewise, in the hair follicle, CRH promotes mast cell degranulation, which also causes perifollicular neuroinflammation [86]. Moreover, an increase in CRH receptors in response to emotional stress has been observed in the skin of patients with alopecia areata [87].

Viral infections, as with other autoimmune diseases [88], have also been implicated in the pathogenesis of alopecia areata, especially infections of Epstein-Barr virus, hepatitis B and C [89], and even recently SARS-CoV-2 [90]. As discussed in previous sections, the activation of IFN-α-producing PDCs by viral infection is an important mechanism that establishes this relationship [50,52]. Supposedly, in a similar mechanobiological manner, cases of alopecia areata have been reported following vaccination [91].

Certain lifestyle factors play a role in the onset and development of alopecia areata [23]. Smokers have a higher risk of developing alopecia than non-smokers [92]. Although the details of the influence of tobacco on the pathogenesis of alopecia areata are unknown, it could cause a Th17-mediated inflammation in the hair follicle [93]. Moreover, sleep quality could also influence the development of alopecia areata, although there are observational studies that have shown that patients with sleep disorders have an increased risk of developing alopecia areata [94], while others have not found a relationship between sleep quality and the development of alopecia areata [95]. Sleep disturbance affects the immune system on several levels and due to the autoimmune nature of alopecia areata, it is possible that it somehow influences its pathogenesis [96]. Furthermore, obesity increases the risk of developing alopecia areata [97], as with other inflammatory skin diseases [98]. A significant change in obesity is the dysregulation of adipokine production. Its dysregulation contributes to chronic low-grade inflammation and affects the modulation of the immune response, intervening in the pathogenesis of various autoimmune diseases, such as alopecia areata [99].

Figure 1 provides a summary of the factors that influence the complex pathogenesis of alopecia areata.

## 3. Alopecia Areata and Microbiota

### 3.1. Hair Follicle/Scalp Microbiota and Alopecia Areata

#### 3.1.1. Role of the Hair Follicle/Scalp Microbiota in the Pathogenesis of Alopecia Areata

The hair follicle contains a variety of bacteria that can reach deeper compartments of the skin [100]. In addition, many bacteria are often organized in biofilms [101] and therefore their study by usual swab sampling may have limitations because it may not be a representative sample of this area. Matard et al., in 2013, reported for the first time the presence of biofilms in the deep zone of scalp hair follicles in both patients with folliculitis decalvans and healthy controls, suggesting their ubiquity [102].

Thus, the hair follicle allows the superficial skin environment and its microbiota to be linked to all skin layers, creating environmental conditions that are highly favorable for persistent bacterial colonization. These favorable environmental conditions are moisture, vascularization, relatively greater protection from ultraviolet light and a more optimal pH for bacterial growth [103]. The establishment of microbiota in the hair follicle is also favored by its IP status.

It is important to emphasize that given the size of the bacteria, access through the infundibular area should be limited to the deep infundibulum and the sebaceous gland. However, the deeper regions of the infundibulum are permeable (epithelium not as tight as the more superficial infundibulum) [104]. In addition, some skin bacteria have enzyme endowments facilitating this penetration and tissue invasion [105].

Given these mentioned conditions, the colonization of the hair follicle is not erratic or random but is regulated and balanced under physiological conditions where the immune system plays an important role. Under normal conditions, therefore, the system that is established is stable, mutually beneficial and under homeostasis control. Communication is established between the bacteria and the underlying hair follicle tissue, and this process is dynamic since both the microbiota influences the immune microenvironment (functionality and composition of the cells of the immune system) and the characteristics of this habitat modulate the composition of the microbiota present [106]. Several mechanisms are involved in the shaping of the hair follicle microbiota. An efficient control system of the healthy hair follicle is the production of antimicrobial peptides (AMPs) such as β-defensin 1 and 2, psoriasin and RNase 7 [104] of different specificities in response to bacterial metabolites that would inhibit pathogenic bacteria. These AMPs are secreted by keratinocytes located in different regions of the hair follicle [107]. This mechanism involves activation of the innate immune system [108]. On the other hand, outer root sheath keratinocytes can also recruit immune system cells in case of excessive colonization or dysbiosis in the hair follicle by secreting cytokines and chemokines [103]. In addition, the connective tissue of the hair follicle is heavily populated with macrophages and mast cells. Bacteria-derived metabolites stimulate keratinocytes, as mentioned above, and these can secrete growth factors for mast cells [109], which are cells with antibacterial capacity, and so they influence the modeling of the microbiota of the hair follicle [110]. In addition, Langerhans cells located mainly in the upper part of the hair follicle (a region where the IP is not yet widely evident) extend their dendrites between the keratinocytes to capture and process bacteria-derived antigens and present them to T-cells. This triggers an adaptive immune response that shapes the composition of the hair follicle microbiota. Therefore, the access of APCs to bacteria has immunomodulatory implications that affect hair follicle homeostasis [111].

The microbiota of the hair follicle under normal conditions in a healthy state is quite similar to that of the skin [105]. There is an abundance of the phylum Actinobacteria, especially of the Propionibacteriaceae and Corynebacteriaceae families, and Firmicutes (*Staphylococcus* and *Streptococcus* genera) and Proteobacteria phylum [103,112]. This healthy hair follicle microbiota prevents colonization by pathogens, stimulates the production of cytokines related to the initiation and maintenance of the immune response [104], and influences the reduction in inflammation and tissue repair [113]. However, there is significant uncertainty about how dysbiosis in the microbiota of the hair follicle affects the growth cycle, regeneration, and immune environment of the hair follicle itself.

The studies detailed below reporting changes in the microbiota of the scalp of patients with alopecia areata do not answer the question of whether these changes are a causal factor or a secondary phenomenon to alopecia areata. However, alterations in the microbiota of the hair follicle or the penetration of material of bacterial origin and immunogenic could influence the modulation of cutaneous immune reactions and inflammatory processes. A peribulbar inflammation is a key process in the pathogenesis of alopecia areata. Moreover, during the anagen phase, the hair follicle is more vulnerable, and growth depends on a strong and robust IP around the bulb [22,38]. In this context, the epithelium around the bulge may be exposed to external (bacteria, metabolites, or bacterial products) or internal pro-inflammatory stimuli (immune system signals triggered by bacteria). Therefore, the microbiota of the hair follicle could be an influential factor in the pathogenesis of alopecia areata.

In relation to this, a higher prevalence of atopic dermatitis has been found in patients with alopecia areata [114]. In the pathogenesis of atopic dermatitis, an important factor is the functionality of T-cells Th17 and Th2, which in turn may be influenced by the skin microbiota [115]. Polak-Witka et al. formulate the hypothesis that the dysbiosis established in alopecia areata and atopic dermatitis could corrupt the IP of the hair follicle and the alteration of the barrier function of the skin in atopic dermatitis could facilitate the penetration of bacterial antigens into deep compartments of the hair follicle [104]. In fact, studies have recently been carried out on the efficacy of drugs that act at the level of the immune mechanisms that are common in alopecia areata and atopic dermatitis [116].

#### 3.1.2. Analysis of the Hair Follicle/Scalp Microbiota in Alopecia Areata Patients

Pinto et al., in 2019, showed for the first time the alterations of the scalp microbiota in patients with alopecia areata compared to healthy controls [117] (Table 1). In a small study with 15 alopecia areata patients and 15 healthy controls, the composition of the microbiota of the scalp surface of alopecia areata lesions obtained by a swab procedure was determined by Next Generation Sequencing (NGS) of the 16S rRNA bacterial gene (universal bacterial taxonomy biomarker). This methodology together with other omics techniques has allowed for an increasingly comprehensive characterization of the microbiome [118]. An increase in α-diversity was observed in alopecia areata patients, which is consistent with other scalp pathologies [119]. The two main phyla in the scalp microbiota in both groups of study subjects were Actinobacteria and Firmicutes with similar abundances. In addition, a qPCR (quantitative polymerase chain reaction) to quantify the presence of *Cutibacterium acnes*, *Staphylococcus aureus* and *Staphylococcus epidermidis* species was also performed. *C. acnes*/*S. epidermidis* and *C. acnes*/*S. aureus* ratios were significantly increased in patients with alopecia areata, which would become these species such as biomarkers of alopecia areata. These results agree with those that have shown that the *Cutibacterium* and *Staphylococcus* genera evidenced reciprocal inhibition in the scalp [120].

Another extended and larger study on scalp microbiota in patients with alopecia areata was published by Pinto’s group in 2020. Pinto et al. analyzed the scalp microbiota in the same way (swab procedure, NGS 16S rRNA gene) as in their 2019 work [117] but in 47 patients with alopecia areata and 47 healthy controls [121] (Table 1). In addition, four patients with alopecia areata and four healthy controls were biopsied, and the microbiota of the dermis, epidermis and hypodermis of the scalp were analyzed. Regarding the biopsies, several differences were found in the different layers of the scalp skin of alopecia areata lesions. At the dermis level, a decrease in *Candidatus Aquiluna*, two operational taxonomic units (OTUs) representing Microthrixaceae family and ACK-M1 and *Staphylococcus* was observed, as well as an increase in *Acinetobacter*. At the epidermis level, an increase in the *Anaerococcus* and *Neisseria* genera was observed, as well as a total absence of the SMB53 genus belonging to the Clostridiaceae family and a decrease in *Staphylococcus* (as in the dermis). *Anaerococcus* has been implicated in stimulating the secretion of AMPs by keratinocytes in other inflammatory skin diseases and could therefore be involved in the pathogenesis of alopecia areata [122]. At the hypodermis level, no significant differences were found in the composition of the microbiota between alopecia patients and controls. The study of α-diversity indices did not find differences between the different skin layers. The authors using tools such as PICRUSt (Phylogenetic Investigation of Communities by Reconstruction of Unobserved States) or KEGG (Kyoto Encyclopedia of Genes and Genomes) that allow us to predict the metabolic pathways involved in the microbiota of the study groups, found some functional profiles that were increased in the patients with alopecia areata such as environmental information processing (bacterial chemotaxis and flagellar assembly) and cellular antigens pathway. This functional capacity related to alopecia areata could be involved in the activation of T-lymphocytes, which play a key role in autoimmune processes [123]. However, this functional analysis failed to discriminate profiles when comparing the microbiota of each skin layer in a paired case-control approach. In addition, a qPCR of genes involved in genetic susceptibility to alopecia areata was performed and correlated with the composition of the microbiota at the genus level. *Anaerococcus*, *Neisseria* and *Acinetobacter* which are increased in alopecia areata patients correlated negatively with the FAS and SOD2 genes but positively with the NOD2 gene. This could imply a close interaction between the host and the microbiota established in alopecia areata patients.

Juhasz et al., in 2020, analyzed through NGS of the 16S rRNA gene the scalp microbiota obtained by swabbing in 25 patients with alopecia areata or alopecia universalis or alopecia totalis and compared it to 25 healthy controls [124] (Table 1). In contrast to the first study by Pinto et al. [117], no significant differences in α-diversity were detected in the two study groups. The beta-diversity study also did not report evidence of clustering of any kind, so the structure of the microbiota of patients with alopecia areata and controls could not present significant dissimilarities overall. However, a paired study of the composition of the different taxonomic groups indicated a significant decrease in the Clostridia class in patients with alopecia areata compared to healthy controls.

Recently, Wong et al., in 2022, also compared the scalp microbiota obtained by swab procedure by NGS of the 16S rRNA gene [125] (Table 1). In 33 alopecia areata patients (26 and 7 with moderate and severe symptoms, respectively) and 12 healthy controls, observed an increase in the α-diversity in the scalp microbiota of alopecia areata lesions. This observation was already reported in the study by Pinto et al. [117] and could be interpreted to suggest that the unhealthy scalp may be susceptible to present increased bacterial colonization. However, the α-diversity did not discriminate against patients with alopecia areata according to their severity. Likewise, the β-diversity assessed by the Bray–Curtis distance did not exhibit significant differences between the study groups (and severity groups in alopecia areata patients) by principal coordinates analysis (PCoA). However, some differences in composition were observed to be of considerable interest. A decrease in Staphylococcaceae and Burkholderiaceae families was observed in alopecia areata patients versus healthy controls and in alopecia areata patients with severe versus moderate forms. *Staphylococcus caprae* abundance was greatly decreased in patients with severe alopecia areata and to a lesser extent in the moderate condition. Interestingly, the *Cutibacterium* species/*S. caprae* ratio was increased from healthy controls (0.97), moderate alopecia areata (2.13) and severe alopecia areata (16.01). Thus, this microbiota-based biomarker could predict severity in patients with alopecia areata and would be the first biomarker associated with the prognosis of this disease. The role of *Cutibacterium acnes* is unknown in the pathogenesis of alopecia areata. Data from this study may suggest that its increased presence could be associated with greater severity. *C. acnes* is a bacterium of crucial importance in skin diseases such as acne vulgaris [27] and even in patients with androgenic alopecia as it has been observed to increase in the scalp microbiota [126]. 

To date, research into treatments for alopecia areata that modulate the scalp or hair follicle microbiota has barely been conducted. Further research on the therapeutic manipulation of skin microbiota must provide evidence as to whether this is a valuable therapeutic option. The only clinical trial published to date involving a microbiota-modulating topical therapy is that of Rinaldi et al., in 2020 [127] (Table 1). Based on the successes and limitations of autologous platelet-rich plasma (PRP) therapy in alopecia areata [128], Rinaldi et al. evaluated the efficacy of a gel containing bioactive peptides that mimicked the action of PRP together with postbiotics in 160 individuals (80 per group). The postbiotic concept includes any substance secreted as a consequence of the metabolic activity of bacteria, which exerts a beneficial effect on the host, either directly or indirectly [129]. This study was a randomized, double-blind, placebo-controlled clinical trial evaluating the efficacy of a cosmetic product containing biomimetic peptides, *Tropaeolum majus* extract and postbiotics such as plantaricin A and *Lactobacillus kunkei* bee bread, in alopecia areata patients with a SALT (severity of alopecia tool) score between S2 and S5. In the group treated with the postbiotic-enriched gel, complete regression was observed in 47% of patients, while 14% experienced partial regression and only 6% did not respond to treatment. On the other hand, in the placebo group, only 5% of alopecia areata patients achieved complete regression. This clinical trial could be an example of the relevance of the skin microbiota in alopecia areata.

### 3.2. Gut Microbiota and Alopecia Areata

#### 3.2.1. Role of the Gut Microbiota in the Pathogenesis of Alopecia Areata

The human gastrointestinal tract harbors 100 trillion bacteria, most of which inhabit the large intestine, including about 1000 species [130]. These bacteria have cooperatively co-evolved with humans. This translates into a 1.3 to 1 ratio of bacteria to human cells, so there are more bacteria than human cells in our organisms. The gut microbiota encodes more than 4 million genes (for approximately 25,000 genes contained in the human genome) and is involved in numerous metabolic reactions that influence the host’s physiology and metabolism to a substantial degree [118]. 

The composition of the gut microbiota tends to be modulated to a greater degree by environmental or lifestyle factors, for example diet or exposure to antibiotics (or other drugs, such as proton pump inhibitors such as omeprazole), rather than by host genetic factors [131]. Western diet, rich in saturated fat and poor in fiber, leads to an unhealthy metabolic profile and markedly influences microbiota composition and diversity [132].

The gut microbiota is stable in healthy adults and consists of a highly adaptive microbial community [133]. A major challenge is to define a “healthy” gut microbiota due to its high gut variability, as it is affected by a variety of known and unknown factors, as well as this, some variability may be stochastic [134]. Despite this high interindividual variability and temporal intraindividual variability in the gut microbiota composition, the metabolic functionality of the gut microbiota is much less diverse and more conserved [133]. This connects with the concept of functional redundancy. It has even been suggested that the functional definition may be superior to the taxonomic definition in distinguishing "abnormal" from "normal" gut microbiota [135]. 

Furthermore, a huge portion of the human body’s immune function is involved in maintaining the balance of the gut microbiota, as 70% of lymphocytes are in gut-associated lymphoid tissue. Given this scenario, it is evident that the gut microbiota is closely involved in the inflammatory response and commensal bacteria act as immune regulators that are critically implicated in the processes of immune tolerance [136]. Disruption of the regulation of these immune responses contributes to the development of inflammatory-based diseases [137]. It has even been observed that changes in the composition and diversity of the gut microbiota can lead to alterations in immunity and inflammation in organs at a distance from the gut [138].

In recent years, the immunomodulatory role of the gut microbiota in distant organs has acquired considerable relevance. Specifically in the gut–skin axis, the gut microbiota basically modulates the functionality and composition of the innate and adaptive immune system, and vice versa [139]. This fact explains why certain skin diseases manifest gut comorbidities [140] and suggests a relationship between the presence of gut dysbiosis and the imbalance of skin homeostasis, with a special role of the gut microbiota in the pathogenesis of several inflammatory skin diseases [141]. On the other hand, although there is still no robust evidence of the role of the gut microbiota in the context of the pathogenesis of alopecia areata, a connection with other autoimmune diseases has been well established [24,142,143]. On this basis, some authors have recently proposed the hypothesis of the participation of the gut microbiota in the pathogenesis of alopecia areata [144]. Indeed, patients with alopecia areata present an increased risk of developing autoimmune diseases [13] and these pathologies could possibly share common pathogenic mechanisms where the gut microbiota could play a key role. 

Preclinical studies in murine models, despite their limitations regarding the translation of results to humans, have provided much relevant information concerning the pathogenesis of alopecia areata [145]. Indirect but significant evidence of a possible relationship between gut microbiota and alopecia areata has been obtained from these murine models. Firstly, C3H/HeJ mice were observed to develop a spontaneous and inherited form of idiopathic inflammatory bowel disease [146], which is also a potential comorbidity that can occur in humans [147] and where the gut microbiota is an important factor in its onset and development [148]. Moreover, normal-haired C3H/HeJ mice grafted with skin from spontaneously alopecia areata-affected mice were fed a diet enriched with soy oil and a high percentage did not develop alopecia areata [149]. This could indicate, apart from the properties of soy oil, the capacity of the diet and indirectly the capacity of the intestinal microbiota, to influence the development of this disease. Nair et al. engrafted skin from alopecia areata-affected C3H/HeJ mice onto unaffected C3H/HeJ mice, which induced the disease in the non-affected C3H/HeJ mice. They had previously treated a group of engrafted mice with a broad-spectrum antibiotic cocktail. The antibiotic-treated mice were protected from developing alopecia areata, but the untreated mice developed hair loss. In addition, a decrease in skin infiltrating T-lymphocytes CD8+NKG2D+ was observed in the antibiotic-treated mice group, indicating a possible role of the gut microbiota in the infiltration of T-cells into the hair follicle that collapses their IP in alopecia areata [150].

On the other hand, an important mechanism in the pathogenesis of alopecia areata is the presence and functionality of T-lymphocytes Treg. Treg are an important and active agent in peripheral tolerance processes, are essential in the prevention of the development of autoimmune diseases [151] and are particularly abundant in hair follicles [152]. In this regard, Scharschmidt et al., in 2017, reported that commensal bacterial colonization of the hair follicle in the postnatal period is involved in the abrupt migration of Treg in the skin (and hair follicle) during a defined postnatal stage [153]. Nevertheless, the role of Treg in the hair follicle, except for the suppressive function of effective T cells, is not fully established. It has been reported that Treg transference blocks the onset of the alopecia areata in C3H/HeJ mice after induction of localized hair loss following T-cells CD8+ injection [154]; therefore, they are important in the pathogenesis. The key could reside in the study by Ali et al., in 2017, who demonstrated that hair follicle Tregs promote regeneration by increasing both the number and differentiation of the stem cells [155]. Recent studies have also indicated that Treg subpopulations in alopecia areata lesions are functionally distinct from those in unaffected tissue and skin from healthy controls [156]. In this regard, how can the gut microbiota influence the pathogenesis of alopecia areata? A gut dysbiosis with depletion of α-diversity and short-chain fatty acid (SFCA) producing bacteria could result in a stressor of the immune system in subjects genetically susceptible to alopecia areata [144]. SFCAs are an influential class of bacteria-derived metabolites from the anaerobic fermentation of complex polysaccharides and oligosaccharides from the diet (dietary fiber), which can directly activate G-coupled receptors, inhibit histone deacetylases, and serve as energy substrates and thus affect various physiological processes [157]. Many studies have supported the influence of SFCAs on the number and functionality of intestinal Treg [158], which may have an impact on peripheral tolerance processes, and thus, on autoimmune diseases such as alopecia areata. A low fiber intake, a decrease in SFCA-producing bacteria or both simultaneously will lead to a decreased intestinal bacterial synthesis of SFCA. In this sense, the gut microbiota could be implicated in the pathogenesis of alopecia areata.

Dysfunction of the intestinal barrier with disruption at the level of the tight junctions (TJs) or damage to the mucosal layer often leads to increased intestinal permeability. This pathological state is known as “leaky gut” syndrome (LGS) [142,159]. LGS initiates an inflammatory response in the intestinal and extraintestinal tissue. Translocation of commensal and pathogenic bacteria occurs, causing a disruption in immune homeostasis and inducing systemic inflammation. Gut dysbiosis leads to mucosal barrier dysfunction and an inflammatory response that predisposes to systemic diseases such as autoimmune diseases [142,159]. Despite the pathogenic similarities between alopecia areata and other autoimmune diseases, there is no evidence of an association with increased intestinal permeability in these patients. Recently, Hacınecipoğlu et al. did not find significant differences in serum zonulin (a TJ protein) levels in 70 alopecia areata patients and 70 healthy controls [160]. However, this possible association should be further examined.

Important evidence for the potential role of gut microbiota modulation in the course of alopecia areata is the anecdotal observations of Rebello [161] and Xie [162], where hair repopulation was observed in alopecia areata patients after undergoing faecal material transplantation (FMT) for other indications such as *Clostridiodes difficile* infection or Crohn’s disease.

#### 3.2.2. Analysis of the Gut Microbiota in Alopecia Areata Patients

Recently, limited studies have characterized the gut microbiota of patients with alopecia areata. All these studies have been based on the NGS of the 16S rRNA bacterial gene in stool samples.

Moreno-Arrones et al., in 2020, studied for the first time the gut microbiota in 15 patients with alopecia universalis and compared it to a group of 15 healthy controls [163] (Table 2). Although no significant differences in α and β-diversity were found between the two groups, using the LefSe (linear discriminant analysis effect size) tool, characteristic increases in alopecia areata patients in Erysipelotrichaceae, Lachnospiraceae and Eggerthellaceae families, and *Holdemania filiformis*, *Parabacteroides johnsonii*, *Clostridiales vadin* BB60 group, *Bacteroides eggerthii*, and *Parabacteroides distasonis* genera were observed. The authors selected biomarker bacteria with LDA scores > 3 and obtained ROC curves, so that those with the highest diagnostic efficacy were *P. distasonis* and *Clostridiales vadin* BB60 group with AUC (area under curve) > 0.7. In addition, these two bacteria were subjected to a multivariate regression model predictive of alopecia areata status and a combined AUC = 0.804 was obtained. This model indicated that a 25% increase in the abundance of *P. distatonis* and *Clostridiales vadin* BB60 group increased the risk of developing alopecia universalis by 9.4% and in the case of *Clostridiales vadin* BB60 by 11.4%. Therefore, this study presents two potential diagnostic biomarkers for alopecia universalis, a severe form of alopecia areata.

Lu et al., in 2021, analyzed and compared the gut microbiota of 33 alopecia areata patients and 35 healthy controls, both groups recruited from Shanghai (China) [164] (Table 2). As in the Moreno-Arrones study [163], no significant differences in α-diversity were observed between the two study groups. The ADONIS tool, which allows permutational multivariate analysis of variance using distance matrices, which in this report was the unweighted UniFrac distance, confirmed that the structure of the microbiota of the alopecia areata patients and healthy controls was significantly different. Based on the LefSe tool, the dysbiosis of patients with alopecia areata was characterized by increases in *Blautia*, *Phyllobacterium*, *Dorea*, *Anaerostipes*, *Megasphaera*, *Collinsella*, *Sphingomonas*, and *Pseudomonas* genera, and in Erysipelotrichaceae family. In addition, a random forest model was applied to obtain three biomarker bacteria to distinguish between patients with alopecia areata and healthy controls, such as *Achromobacter*, *Megasphaera* and *Lachnospiraceae Incertae Sedis*. These biomarkers could be useful in the early diagnosis of the disease as well as a therapeutic target. The discrepancies in the differential potential biomarkers between the findings of Moreno-Arrones [163] and Lu [164] indicate the importance of the geographical location as a confounding factor since both studies were conducted in regions where the diet is totally different and this could explain, at least partially, the variability observed. This is a critical issue to be considered for future research.

Rangu et al., in 2021, investigated for the first time the gut microbiota in patients with alopecia areata in a paediatric population in a study with 21 children and their siblings (as a control group) aged 4 to 17 years [165] (Table 2). In this project, the gut microbiota was analyzed using a shotgun metagenomics methodology. As in previous studies [163,164], there were no significant differences at the α-diversity level between the two study groups. Moreover, there were no differences at the β-diversity level, as the Bray–Curtis distance analysis did not indicate any significant differences. However, a linear mixed model revealed that *Ruminicoccus bicirculans* exhibited lower relative abundance in alopecia areata patients. Therefore, in terms of composition and diversity, the differences between the two study groups are nearly minimal. Nevertheless, when analyzing the functionality of the gut microbiota, that is, the gene function, 20 genes were detected in a different status in children with alopecia areata compared to their siblings. Thus, a decrease in two spore germination genes and an increase in two metal transportation genes and one multidrug resistance gene in alopecia areata patients were discovered, among others. The authors highlighted that the transition of the gut microbiota from childhood to adulthood could be important for understanding microbiota-host interactions in relation to alopecia areata.

An important consideration when interpreting the results of studies exploring the gut microbiota in patients with alopecia areata is the concomitant treatment. In this way, oral corticosteroids can modify the gut microbiota [166], so these studies should emphasize whether patients are on treatment.

To date and to our knowledge there are no published clinical trials of probiotics as an adjuvant therapy in alopecia areata, although as mentioned above there are published clinical cases demonstrating the success of FMT [161,162]. Several probiotics have shown beneficial effects as adjuvant treatments in some inflammatory skin diseases such as atopic dermatitis, psoriasis, acne, etc. [26,27]. Given the exposed nature of the pathogenesis of alopecia areata and given the immunomodulatory or anti-inflammatory properties of some probiotics [167], it would be convenient in the future to explore this research.

## 4. Conclusions

Alopecia areata is an autoimmune-based disease with a complex and multifactorial pathogenesis. The hair follicle (scalp) and gut microbiota could be involved in the onset and development of this disease. This narrative review has examined the published literature in this regard and, although there is evidence of abnormalities in the microbiome of patients with alopecia areata, it is not yet strongly supported. Furthermore, it is unknown whether these changes are a cause or a consequence of the disease. However, current evidence suggests that the hair follicle (scalp) and gut microbiota could be involved, at least at some level, in the pathogenesis of alopecia areata. Further research should be conducted mainly through descriptive studies in humans and through preclinical models to obtain improved evidence. This would provide a better understanding of how to modulate the microbiome through probiotics in a targeted and precise manner, in order to obtain an adjuvant therapy that could reduce the adverse effects of the current therapy and improve the clinical course of the disease.

## Figures and Tables

**Figure 1 genes-13-01860-f001:**
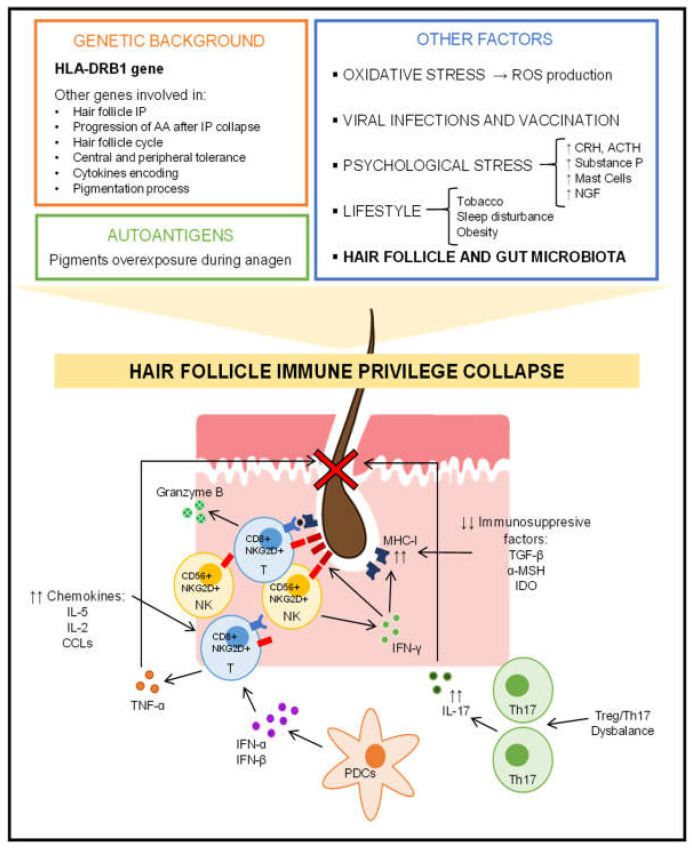
Overview of the pathogenesis of alopecia areata. IP: Immune privilege; AA: Alopecia Areata; ROS: Reactive Oxygen Species; CRH: Corticotropin-Releasing Hormone; ACTH: Adrenocorticotropic Hormone; NGF: Nerve Growth Factor; IL-2: Interleukin 2: IL-5: Interleukin 5; IL-17: Interleukin 17; CXCLs: CXC Ligands; TNF-α: Tumor Necrosis Factor α; INF-α: Interferon α; INF-β: Interferon β. INF-γ: Interferon γ; PDCs: Plasmacytoid dendritic cells; NK: Natural Killers; Th17: T-cells Th17; Treg: T-cells Treg; MHC-I: Major Histocompatibility Complex Class I; NKGD2D: NK Cell Receptor D; TGF-β: Transforming Growth Factor β; α-MSH: α-Melanocyte-Stimulating Hormone; IDO: Indoleamine-2,3-Dioxygenase.

**Table 1 genes-13-01860-t001:** Main studies scalp microbiota-related in patients with alopecia areata. NGS: Next Generation Sequencing; rRNA: Ribosomal Ribonucleic Acid; qPCR: Quantitative Polymerase Chain Reaction; OTU: Operational Taxonomic Unit; PICRUSt; Phylogenetic Investigation of Communities by Re-construction of Unobserved States; KEGG: Kyoto Encyclopedia of Genes and Genomes; PCoA: Principal Coordinates Analysis; SALT: Severity of Alopecia Tool.

Reference	Methodology and Study Population	Key Results
Pinto 2019 [117]	15 patients with alopecia areata and 15 healthy controls. Study of the microbiota of the scalp surface of alopecia areata lesions by swab procedure. NGS 16S rRNA. qPCR of *Cutibacterium acnes*, *Staphylococcus aureus* and *Staphylococcus epidermidis*.	Increase in α-diversity in alopecia areata patients. Actinobacteria y Firmicutes are de the two main phyla in the scalp microbiota in both study groups (similar abundances). *C. acnes*/*S. epidermidis* and *C. acnes*/*S. aureus* ratios significantly were increased in alopecia areata patients.
Pinto 2020 [121]	47 patients with alopecia areata and 47 healthy controls. Study of the microbiota of the scalp surface of alopecia areata lesions by swab procedure. Study of the microbiota of the dermis, epidermis, and hypodermis of the scalp by biopsy in four patients with alopecia areata and four healthy controls. NGS 16S rRNA. qPCR of genes involved in genetic susceptibility to alopecia areata.	Biopsies: In alopecia areata patients at the dermis level it was observed a decrease in *Candidatus Aquiluna*, *Staphylococcus* and 2 OTUs representing Microthrixaceae and ACK-M1 families, and an increase in *Acinetobacter*. In alopecia areata patients at the epidermis level, an increase in *Anaerococcus* and *Neisseria* was observed, and an absence of SMB53 genus (Clostridiacea), and a decrease in *Staphylococcus*. No significant differences were reported at the hypodermis level between the two study groups. No differences in α-diversity between the skin layers. PICRUSt, KEGG: In alopecia areata patients it was observed an increase in some functional profiles such as the environmental information processing and the cellular antigen pathway. *Anaerococcus*, *Neisseria* and *Acinetobacter* correlated negatively, in alopecia areata patients, with FAS and SOD2 genes but positively with the NOD2 gene.
Juhasz 2020 [124]	25 patients with alopecia areata, alopecia universalis or alopecia totalis and 15 healthy controls. Study of the microbiota of the scalp surface of alopecia areata lesions by swab procedure. NGS 16S rRNA.	No significant differences were detected in α-diversity between the two study groups. Β-diversity analysis did not show any differences between the two study groups. Significant decrease in Clostridia in alopecia areata patients.
Wong 2022 [125]	33 patients with alopecia areata (26 and 7 with moderate and severe symptoms, respectively) and 12 healthy controls. Study of the microbiota of the scalp surface of alopecia areata lesions by swab procedure. NGS 16S rRNA.	Increase in α-diversity in alopecia areata patients. A-diversity did not discriminate against patients with alopecia areata according to their severity. PCoA using Bray-Curtis distance did not exhibit significant differences between the study groups (and severity groups in alopecia areata patients). Decrease in Staphylococcaceae and Burkholceriaceae in alopecia areata patients and in severe versus moderate forms. Strong decrease in *Staphylococcus caprae* in patients with severe alopecia areata. *Cutibacterium* species/*S. caprae* ratio was increased from healthy controls (0.97) to moderate alopecia areata (2.13) and severe alopecia areata (16.01).
Rinaldi 2020 [127]	Randomised, double-bind, placebo-controlled, clinical trial. Evaluation of the efficacy of a gel containing biomimetic peptides, *Tropaeoleum majus* extract and postbiotics such as plantaricin A and *Lactobacillus kunkei* bee bread in 160 alopecia areata patients with a SALT score between S2 and S5.	In the postbiotic-enriched gel group, were reported a “complete regression” in 47% of patients, a “partial regression” in 14% of patients and a “no respond” in 6% of patients. In the placebo group, only 5% of patients achieved “complete regression”.

**Table 2 genes-13-01860-t002:** Main gut microbiota-related studies in patients with alopecia areata. FMT: Faecal Material Trans-plantation; CDI: Clostridiodes difficile Infection; NGS: Next Generation Sequencing; rRNA: Ribo-somal Ribonucleic Acid; LefSe: Linear Discriminant Analysis Effect Size; AUC: Area Under Curve.

Reference	Methodology and Study Population	Key Results
Rebello 2017 [161]	Case Report. two patients with *Clostridiodes difficile infection* and treated with FMT. Moreover, both patients presented alopecia areata.	This report highlights two patients with coexisting alopecia areata and recurrent CDI who experienced hair regrowth after FMT. This could suggest a strong immunological response to FMT.
Xie 2019 [162]	Case Report. A patient diagnosed with noninfectious diarrhea, depressive disorder, and patchy alopecia areata. The patient was treated with six rounds of FMT.	It was observed new hair growth on the affected region of his scalp without taking any other therapies for alopecia areata before and after FMT.
Moreno-Arrones 2020 [163]	15 patients with alopecia universalis and 15 healthy controls. Stool samples. NGS 16S rRNA.	No significant differences in α-diversity between the two study groups. β-diversity analysis did not show any differences between the two study groups. Using the LefSe tool it was observed characteristic increases in alopecia areata patients in Erysipelotrichaceae, Lachnospiraceae and Eggerthellaceae families, and *Holdemania filiformis*, *Parabacteroides johnsonii*, *Clostridiales vadin* BB60 group, *Bacteroides eggerthii*, and *Parabacteroides distasonis* genera. The highest diagnostic efficacy was reported for *P. distasonis* and *Clostridiales vadin* BB60 group with AUC > 0.7. These bacteria were subjected to a multivariate regression model predictive of alopecia areata status and a combined AUC = 0.804 was obtained. This model indicated that a 25% increase in the abundance of *P. distatonis* and *Clostridiales vadin* BB60 group increased the risk of developing alopecia universalis by 9.4% and in the case of *Clostridiales vadin* BB60 by 11.4%.
Lu 2021 [164]	33 patients with alopecia areata and 30 healthy controls. Stool samples. NGS 16S rRNA.	No significant differences in α-diversity between the two study groups. ADONIS tool using UniFrac distance exhibit significant differences between the study groups. Using the LefSe tool it was observed characteristic increases in alopecia areata patients in *Blautia*, *Phyllobacterium*, *Dorea*, *Anaerostipes*, *Megasphaera*, *Collinsella*, *Sphingomonas*, and *Pseudomonas* genera, and in Erysipelotrichaceae family. *Achromobacter*, *Megasphaera* and *Lachnospiraceae Incertae Sedis* were identified by a random forest model as biomarker bacteria for alopecia areata status.
Rangu 2021 [165]	21 children with alopecia areata aged 4 to 17 years and their siblings (as a control group) aged 4 to 17 years. Stool samples. Shotgun Metagenomics.	No significant differences in α-diversity between the two study groups. β-diversity analysis using the Bray–Curtis distance did not show any differences between the two study groups. A linear mixed model revealed that *Ruminicoccus bicirculans* exhibited lower relative abundance in alopecia areata patients.

## Data Availability

Not applicable.

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
