# Peer review of "How Our Microbiome Influences the Pathogenesis of Alopecia Areata"

_genes, 2022, doi:10.3390/genes13101860_

Round 1
Reviewer 1 Report
The aim of this narrative review is to examine the published literature on the cutaneous and intestinal microbiota in 20 alopecia areata to be able to establish a pathogenic link.
As cutibacterium spp is consider as an interesting bacteria for healhy scalp in previous paper especially when we compare to dandruff condition. How we may explain that here Cutibacterium may turn the bad guy.
for the paper is really when written and very interesting.
I will accept but would like to have a comment about my observation
thanks
Author Response
Dear Reviewer, thank you for your comment and I try to answer here:
There are few studies and data on the follicular microbiome; however, it is documented that Actinobacteria, Firmicutes and Proteobacteria (Lousada et al. Exploring the human hair follicle microbiome. Br J Dermatol 2021; 184(5): 802-15) predominate at the scalp level, with the genus Cutibacterium spp. being the most frequent and Cutibacterium acnes as the most prevalent bacterium togheter with Staphylococcus spp. (with a predominance of Staphylococcus epidermidis).
A study that determined the microbiota in healthy patients and subjects with alopecia areata found the following data of bacteria at the scalp level: Actinobacteria (56.3% control vs. 57.4% in AA), Firmicutes (35.2% control vs. 29.2% in AA), Cutinobacterium including C. acnes (45.6% control vs. 55.1% in AA), Staphylococcus (32.6% control vs. 27.4% in in AA). Likewise, in patients with alopecia areata, was observed an increase in the C. acnes/S. epidermidis C. acnes/S. aureus ratio (Pinto et al. Scalp bacterial shift in Alopecia areata. PLoS One 2019; 14(4): e0215206).
The role of C. acnes is unclear, as it is a ubiquitous bacteria in healthy human skin (Byrd et al. The human skin microbiome. Nat Rev Microbiol 2018; 16(3): 143-55). Therefore, while C. acnes is a predominant bacteria in sebaceous regions of the skin, with a vital role in cutaneous homeostasis, even acting in the prevention of pathogens (Christensen et al. Bacterial skin commensals and their role as host guardians. Benef Microbes 2014; 5(2): 201-15) it can also act as an opportunistic pathogen (Sánchez-Pellicer et al. Acne, Microbiome, and Probiotics: The Gut-Skin Axis. Microorganisms 2022; 10(7): 1303).
Tomida et al. sequenced the complete genome of 82 strains of C. acnes and observed that it had a constant core region and a variable region. This variability would explain its commensal or pathogenic phenotype (Tomida et al. Pan-genome and comparative genome analyses of propionibacterium acnes reveal its genomic diversity in the healthy and diseased human skin microbiome. mBio 2013; 4(3): e00003-13).
Therefore, the role of C. acnes in the pathogenesis of alopecia areata is unknown, although some studies have reported its presence as a potential biomarker. The fact that it is a bacteria normally present in the cutaneous microbiota and that in some cases it can behave as a pathogen, will depend to a large extent on its genetic characteristics.
Reviewer 2 Report
Dear Authors,
I read with interest your paper. It is an extensive review. Moreover, as novel treatments are being developed for alopecia areata (AA), the subject matter and content of the review are of practical interest at the present time Although the text is useful, I would suggest some changes:
1) Introduction: I would remove some paragraphs, as the introduction is too long. Paragraphs from line 25 to 79 could be removed, as they focus too much on issues that are irrelevant to the objective of the review.
2) Introduction, treatment section: Although there is a lack of evidence, there are novel treatments, such as Baricitinib which have been recently approved. I would suggest changing the paragraph to make it more easily understandable -> It could be structured as follows: Mild cases of AA are treated with topical or intralesional corticosteroids; Moderate to severe cases of AA are treated with systemic corticosteroids, topical contact therapy, immunosuppressive agents or JAK inhibitors.
3) Introduction, treatment section: some specific references to the current treatment options for AA should be referenced, such as oral corticosteroid treatment (Sánchez-Díaz M, Montero-Vilchez T, Bueno-Rodriguez A, Molina-Leyva A, Arias-Santiago S. Alopecia Areata and Dexamethasone Mini-Pulse Therapy, A Prospective Cohort: Real World Evidence and Factors Related to Successful Response. J Clin Med. 2022 Mar 18;11(6):1694. doi: 10.3390/jcm11061694. PMID: 35330017; PMCID: PMC8949115), or Janus Kinase inhibitors, baricitinib, which has been approved, and tofacitinib (King B, Ohyama M, Kwon O, Zlotogorski A, Ko J, Mesinkovska NA, Hordinsky M, Dutronc Y, Wu WS, McCollam J, Chiasserini C, Yu G, Stanley S, Holzwarth K, DeLozier AM, Sinclair R; BRAVE-AA Investigators. Two Phase 3 Trials of Baricitinib for Alopecia Areata. N Engl J Med. 2022 May 5;386(18):1687-1699. doi: 10.1056/NEJMoa2110343. Epub 2022 Mar 26. PMID: 35334197; Sanchez-Diaz M, Diaz-Calvillo P, Rodriguez-Pozo JA, Tercedor-Sánchez J, Cantudo-Cuenca MR, Molina-Leyva A, Arias-Santiago S. Tofacitinib for Treatment of Alopecia Areata: Real-world Evidence and Factors Associated with Therapeutic Response. Acta Derm Venereol. 2022 Jun 15;102:adv00736. doi: 10.2340/actadv.v102.2036. PMID: 35578820 ).
4) It is mandatory to include a Material and Methods section, to explain how the review was conducted (which was the criteria to include the studies, how each study was reviewed, which databases were consulted, which time period was included in the review...).
5) The main body text and the conclusions are well written and provide very detailed information. Moreover, the images and tables are very useful.
6) I would suggest including some information about how most used treatments (oral corticosteroids, JAK inhibitors) could interact with microbiome in AA patients.
Author Response
Reviewer 1 Report.
Dear Authors, I read with interest your paper. It is an extensive review. Moreover, as novel treatments are being developed for alopecia areata (AA), the subject matter and content of the review are of practical interest at the present time. Although the text is useful, I would suggest some changes:
The authors appreciate the comments of the reviewer and then go on to develop the improvements provided.
1) Introduction: I would remove some paragraphs, as the introduction is too long. Paragraphs from line 25 to 79 could be removed, as they focus too much on issues that are irrelevant to the objective of the review.
We assume that the reviewer is really referring to the paragraph from line 64 to 79. We agree with the reviewer that this paragraph does not provide relevant information with respect to the central topic of the review, and we remove it.
2) Introduction, treatment section: Although there is a lack of evidence, there are novel treatments, such as Baricitinib which have been recently approved. I would suggest changing the paragraph to make it more easily understandable -> It could be structured as follows: Mild cases of AA are treated with topical or intralesional corticosteroids; Moderate to severe cases of AA are treated with systemic corticosteroids, topical contact therapy, immunosuppressive agents or JAK inhibitors.
In agreement with the reviewer's suggestion, we have modified one sentence of this paragraph following his indications:
“In relation to this, it could be pragmatically assumed that in clinical practice, mild cases of alopecia areata are treated with topical or intralesional corticosteroids and moderate to severe cases with systemic corticosteroids, immunosuppressive agents, janus kinase (JAK) inhibitors, different types of immunotherapy and biological drugs, minoxidil, laser therapy, etc”.
Also, in the same paragraph, we mention the recent approval of Baricitinib:
“However, research for more effective treatment of alopecia areata continues. In this regard, Baricitinib has recently been approved in both Europe and the United States. It is an oral inhibitor of JAK 1 and 2 that can interrupt the signaling of cytokines involved in the pathogenesis of alopecia areata.”
3) Introduction, treatment section: some specific references to the current treatment options for AA should be referenced, such as oral corticosteroid treatment (Sánchez-Díaz M, Montero-Vilchez T, Bueno-Rodriguez A, Molina-Leyva A, Arias-Santiago S. Alopecia Areata and Dexamethasone Mini-Pulse Therapy, A Prospective Cohort: Real World Evidence and Factors Related to Successful Response. J Clin Med. 2022 Mar 18;11(6):1694), or Janus Kinase inhibitors, baricitinib, which has been approved, and tofacitinib (King B, Ohyama M, Kwon O, Zlotogorski A, Ko J, Mesinkovska NA, Hordinsky M, Dutronc Y, Wu WS, McCollam J, Chiasserini C, Yu G, Stanley S, Holzwarth K, DeLozier AM, Sinclair R; BRAVE-AA Investigators. Two Phase 3 Trials of Baricitinib for Alopecia Areata. N Engl J Med. 2022 May 5;386(18):1687-1699; Sanchez-Diaz M, Diaz-Calvillo P, Rodriguez-Pozo JA, Tercedor-Sánchez J, Cantudo-Cuenca MR, Molina-Leyva A, Arias-Santiago S. Tofacitinib for Treatment of Alopecia Areata: Real-world Evidence and Factors Associated with Therapeutic Response. Acta Derm Venereol. 2022 Jun 15;102:adv00736).
In agreement with the reviewer, including the following references:
- King, B.; Ohyama, M.; Kwon, O.; Zlotogorski, A.; Ko, J.; Mesinkovska, N.A.; Hordinsky, M.; Dutronc, Y.; Wu, W.S.; McCollam, J.; et al. Two Phase 3 Trials of Baricitinib for Alopecia Areata. N Engl J Med 2022, 386(18), 1687-1699. doi: 10.1056/NEJMoa2110343.
- Sánchez-Díaz, M.; Montero-Vilchez, T.; Bueno-Rodriguez, A.; Molina-Leyva, A.; Arias-Santiago, S. Alopecia Areata and Dexamethasone Mini-Pulse Therapy, A Prospective Cohort: Real World Evidence and Factors Related to Successful Response. J Clin Med 2022, 11(6), 1694. doi: 10.3390/jcm11061694.
We would like to thank the reviewer for the reference to Tofacitinib. However, this paragraph introduces the treatment of alopecia areata but it is not aimed at going into detail. We do not think that the reference to Tofacitinib is necessary.
4) It is mandatory to include a Material and Methods section, to explain how the review was conducted (which was the criteria to include the studies, how each study was reviewed, which databases were consulted, which time period was included in the review...).
Dear reviewer, since it is a narrative review (as specified in the objectives, at the end of the introduction), we consider that it would not be appropriate to develop the methodology since it is not a systematic review or a scoping review.
Through this approach, the authors address the most important points we consider to be relevant to the topic in question, without the rigidity of a systematic review that answers a single question by using an established methodology. This is reflected in the last paragraph of the introduction.
5) The main body text and the conclusions are well written and provide very detailed information. Moreover, the images and tables are very useful.
The authors appreciate the comments of the reviewer.
6) I would suggest including some information about how most used treatments (oral corticosteroids, JAK inhibitors) could interact with microbiome in AA patients.
We are grateful to the reviewer for this comment. We certainly had not thought about this point, but it is true that oral corticosteroids can modify the microbiota (there is literature on this matter) and this could be a confounding factor when interpreting the results in a study analysing the gut microbiota of patients with alopecia areata treated with this medication.
We have not found much information on how JAK inhibitors could influence the gut microbiota. Collota et al, in 2020, report that "Treatment of high-fat and high-sugar diet mice with baricitinib did not change diet-induced alterations in the gut, but restored insulin signalling in liver and skeletal muscle, resulting in improvements in diet-induced myosteatosis, mesangial expansion and associated proteinuria" [Collotta D. Baricitinib counteracts metaflammation, thus protecting against diet-induced metabolic abnormalities in mice. Mol Metab 2020; 39: 101009]. This result is noteworthy but open to various interpretations. We do not think it matches with this proposed review.
Reviewer 3 Report
The review article entitled “How our microbiome influences the pathogenesis of Alopecia Areata: A narrative review“ submitted by Pellicer et al. gained my attraction. The paper nicely summarizes the influence of gut microbiota in the context of alopecia areata.
The article was presented in simpler form but looks a little lengthy. If possible, try to shorten the introduction part by putting only interesting facts on the subject. Also, some typos such as eference in line 818 should be corrected. Table titles should be at the top of the Table not at the bottom. I would suggest it to correct it. Future directions if added at the last would give more meaningful conclusions to the work.
Author Response
Reviewer 2 Report.
The review article entitled “How our microbiome influences the pathogenesis of Alopecia Areata: A narrative review submitted by Pellicer et al. gained my attraction. The paper nicely summarizes the influence of gut microbiota in the context of alopecia areata.
The authors are grateful for the comments of the reviewer.
The article was presented in simpler form but looks a little lengthy. If possible, try to shorten the introduction part by putting only interesting facts on the subject.
The authors have considered your comments and we thank you for them. In agreement with reviewer 1, we have also removed a paragraph from the introduction referring to the trichoscopic diagnosis of alopecia areata. However, we find the rest of the paragraphs interesting and introductory to the concepts subsequently developed.
Also, some typos such as reference in line 818 should be corrected.
We change the typographical error "eference" to "Reference". We reviewed the entire manuscript for similar errors
Table titles should be at the top of the Table not at the bottom. I would suggest it to correct it.
According to your suggestion we place the title of the tables and figures at the top.
Future directions if added at the last would give more meaningful conclusions to the work.
Thank you very much for the comment, in the last paragraph of the conclusions we already mentioned the following:
“Further research should be conducted mainly through descriptive studies in humans and through preclinical models to obtain improved evidence. This would provide a better understanding of how to modulate the microbiome through probiotics in a targeted and precise manner, in order to obtain an adjuvant therapy that could reduce the adverse effects of the current therapy and improve the clinical course of the disease.”